# Effects of pH/pCO₂ fluctuations on photosynthesis and fatty acid composition of two marine diatoms, with reference to consequence of coastal acidification

Yu Shang[1], Jingmin Qiu[1], Yuxi Weng[1], Xin Wang[1], Di Zhang[2], Yuwei Zhou[1], Juntian Xu[1,3], Futian Li[1,3]

[1] Jiangsu Key Laboratory of Marine Bioresources and Environment, Jiangsu Ocean University, Lianyungang, 222000, China
[2] School of Ocean, Yantai University, Yantai, 264000, China
[3] Co-Innovation Center of Jiangsu Marine Bio-industry Technology, Jiangsu Ocean University, Lianyungang, 222000, China

*Correspondence to*: Futian Li (futianli@jou.edu.cn)

**Abstract.** Coastal waters are impacted by a range of natural and anthropogenic factors, which superimpose on effects of increasing atmospheric $CO_2$, resulting in dynamically changing seawater carbonate chemistry. Research on the influences of dynamic pH/pCO₂ on marine ecosystems is still in its infancy, although effects of ocean acidification have been extensively studied. In the present study, we manipulated the culturing pH to investigate physiological performance and fatty acid (FA) composition of two coastal diatoms *Skeletonema costatum* and *Thalassiosira weissflogii* in both steady and fluctuating pH regimes. Generally, seawater acidification and pH variability showed neutral or positive effects on specific growth rate, chlorophyll *a*, and biogenic silica contents of two species. Decreased pH inhibited net photosynthetic rate by 27% and enhanced mitochondrial respiration rate of *S. costatum* by 36% in the steady pH regime, while these rates were unaltered by decreased pH in the fluctuating regime. Acidification conditions lead to lower saturated FA and higher polyunsaturated FA proportions in both species, regardless of steady and fluctuating regimes. Our results indicate that coastal acidification could affect primary production in a different way from ocean acidification. Together with the altered nutritional quality of prey for higher trophic levels, coastal acidification might have far-reaching consequence for marine ecosystem functioning.

## 1 Introduction

Carbonate chemistry of coastal waters is impacted by biological metabolism, tidal cycles, upwelling, wind, and terrestrial nutrient inputs, in addition to the dissolution of atmospheric $CO_2$ (Carstensen and Duarte, 2019; Duarte et al., 2013; Kapsenberg and Cyronak, 2019). This results in dynamic changes in carbonate chemistry parameters such as pH and pCO₂ (García-Ibáñez et al., 2024). The amplitude of pH changes in coastal regions could be greater than 1 unit within 24 hours (Duarte et al., 2013), which is larger than the average expected change of 0.3 units by the end of this century (Gattuso et al., 2015). Thus, short-term pH fluctuations will superimpose on the downward trend of pH in coastal waters in the context of ocean acidification. These fluctuations may have potential impacts on marine organisms at different trophic levels, as suggested by limited research (Li et al., 2021; Raven et al., 2020; Schaum et al., 2016; Wahl et al., 2018).

Diatoms are usually one of the dominant phytoplankton taxa in coastal waters, where they contribute to a large proportion of primary production (Tréguer et al., 2018). Coastal diatoms are characterized by high tolerance to dynamic changes in abiotic factors (Key et al., 2010; Li et al., 2016; Strzepek and Harrison, 2004). This tolerance is supported by their special cell structure or fast acclimation rate and broad ecological niche (Armbrust, 2009). The pH tolerance of diatoms varies among species, with some capable of adapting to a wide range of pH levels and exhibiting positive growth (Hansen, 2002; Hinga, 2002). Our previous study found that coastal diatom species may benefit from or be tolerant to diurnal pH fluctuation (Li et al., 2016; Shang et al., 2024). This phenomenon is suggested to be related to the larger pH differences experienced by larger-sized coastal phytoplankton cells between the diffusion boundary layer (DBL) surrounding the cells and the bulk seawater (Flynn et al., 2012). In addition, coastal waters are characterized by large amplitude and high-frequency fluctuations in seawater carbonate chemistry parameters, particularly pH and $pCO_2$ (Duarte et al., 2013). Thus, larger-sized phytoplankton dwelling in coastal waters with turbulent condition should show high tolerance to changes in carbonate chemistry.

Effects of ocean acidification and underlying mechanisms have been extensively studied at different trophic levels on both short- and long-term timescales (Doney et al., 2020; Hancock et al., 2020), yet research on effects of dynamic pH is still in its infancy. Ocean acidification might have various effects on diatoms based on simulated laboratory and field studies, and other environmental drivers could mediate the effects (Gao and Campbell, 2014). These studies are important to reveal the comprehensive consequence of ocean acidification. However, the impacts of fluctuating carbonate chemistry might differ from those of decreased pH and increased $CO_2$ in steady regimes.

Limited studies have focused on how marine phytoplankton perform under fluctuating pH/$pCO_2$ condition, leaving the effects of coastal acidification poorly understood. This knowledge gap impedes accurate predictions of acidification impacts in coastal regions. To capture more details in fluctuating pH regime, we manipulated the culturing pH/$pCO_2$ in a stepwise way by adjusting $pCO_2$ aerated into cultures in the present study, and each step lasts for 24 h. This enabled us to investigate cell performance at each pH level in the fluctuating regime, besides the overall responses. We hypothesize that coastal diatoms could tolerate environmental pH fluctuation, given the dynamic carbonate chemistry in coastal waters and the unignorable pH difference between the DBL of cells and bulk seawater.

## 2 Materials and Methods

### 2.1 Culture conditions and experiment setup

Two typical diatoms *Skeletonema costatum* (originally isolated from coastal waters of Gaogong Island, Jiangsu Province, China) and *Thalassiosira weissflogii* (CCMA 102, originally isolated from Daya Bay, Guangdong Province, China) were cultured in polycarbonate bottles with 500 ml sterile artificial seawater (Sunda et al., 2005). Nutrients were added according to F/2 recipe (Guillard and Ryther, 1962) to ensure cells were not limited by nutrients. Triplicate cultures were set for each treatment and they were cultured in one incubator with light intensity of 150 μmol photons $m^{-2}$ $s^{-1}$ and 12:12 h light and dark

cycle. Culturing temperature was set at 20 °C, which is in the optimal temperature range for growth of two species. Cultures were diluted every three or four days to make sure cells were in the exponential phase, with maximum cell densities below 160,000 and 10,000 cells $ml^{-1}$ for *S. costatum* and *T. weissflogii*, respectively.

To compare effects of pH level and variability on two diatoms, four $pH/pCO_2$ treatments were set: 1) steady ambient $pH/pCO_2$ level (LCs); 2) steady future $pH/pCO_2$ level (HCs); 3) fluctuating ambient $pH/pCO_2$ level with similar mean values of $pH/pCO_2$ with those in LCs treatment (LCf); 4) fluctuating future $pH/pCO_2$ level with similar mean values of $pH/pCO_2$ with those in HCs treatment (HCf) (see Fig.1). The $pH_{NBS}$ values were measured using a pH meter (FE20, Mettler Toledo) with the NBS buffer system and three-point calibration. The $CO_2$ partial pressure of the aerating air was measured with a

$CO_2$ detector (GM70, Vaisala Oyj). To verify the seawater carbonate chemistry parameters, total alkalinity (TA) was also determined. For TA measurement, samples were filtered through cellulose acetate membranes (0.45 μm, Xinya) and determined using the pH method after Anderson and Robinson (1946). The remaining carbonate chemistry parameters were calculated using the CO2SYS program, based on pH and TA (see Table 1), with the carbonic acid dissociation constants from Mehrbach et al. (1973), refitted by Dickson and Millero (1987), and those for sulfuric acid from Dickson (1990). LCs

and HCs cultures were aerated with ambient air and $CO_2$-enriched air, respectively. The $CO_2$-enriched air was achieved by mixing air and $CO_2$ with a $CO_2$ Enricher (CE100, Ruihua). The target $pCO_2$ level (1000 μatm) for HCs cultures was set according to the projected range in the sixth assessment report of the Intergovernmental Panel on Climate Change (Lee et al., 2021). For fluctuating regimes, the $pCO_2$ of aerating air was adjusted every 24 h in a stepwise way. The $pCO_2$ was set as follows: 400-280-400-1000-400 μatm for LCf and 1000-400-1000-1750-1000 μatm for HCf treatments, and each step lasted

for 24 h. This resulted in pH ranging from 7.85 to 8.35 and from 7.6 to 8.1 under LCf and HCf conditions, respectively (Fig. 1). The amplitude and frequency of fluctuations were set based on reported habitat conditions and experimental feasibility. The aerating rate was controlled at 100 ml $min^{-1}$ by a gas flowmeter, and filter units (SLGPR33RB, Millipore) were used to sterilize the aerating air. Cultures were acclimated to four treatments for at least 10 days (i.e. two pH variation cycles for fluctuating regime) before following parameters were measured.

Specific growth rates in fluctuating regimes were measured four times, covering each pH period at 24 h intervals. For comparison, the same sampling frequency was applied to cells in steady regimes. Net photosynthetic and mitochondrial respiration rates were determined during each pH period in fluctuating regimes, and photosynthetic rates in steady regimes were measured twice on different days to monitor potential variations over time. Other parameters were assessed when pH levels in fluctuating regimes matched those of the corresponding steady regimes. Fatty acid compositions were analyzed at

the end of the experiment due to the high biomass requirement.

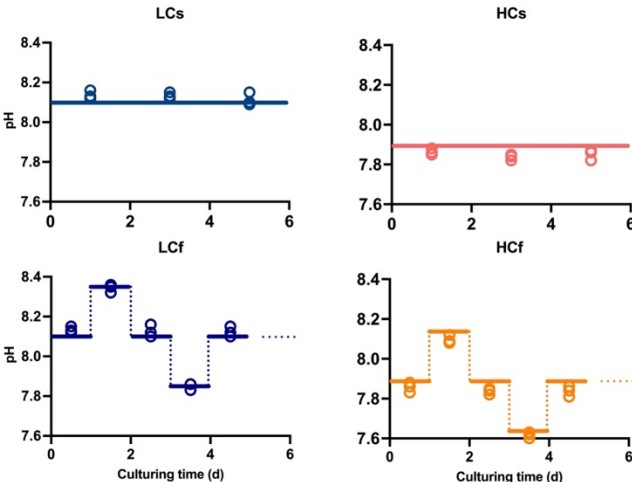

Figure 1. Target (lines) and measured culturing pH (open circles) of steady ambient pH/pCO₂ (LCs), fluctuating ambient pH/pCO₂ (LCf), steady future pH/pCO₂ (HCs), and fluctuating future pH/pCO₂ (HCf) treatments.

Table 1. Carbonate chemistry parameters of steady ambient pH/pCO₂ (LCs), fluctuating ambient pH/pCO₂ (LCf), steady future pH/pCO₂ (HCs), and fluctuating future pH/pCO₂ (HCf) treatments. The unit for TA, dissolved inorganic carbon (DIC), bicarbonate, carbonate, and dissolved $CO_2$ is µmol kg⁻¹.

|  | $pH_{NBS}$ | TA | DIC | $HCO_3^-$ | $CO_3^{2-}$ | $CO_2$ |
|---|---|---|---|---|---|---|
| LCs | 8.14±0.02 | 2271±65 | 1994±71 | 1812±71 | 168±2 | 14±1 |
| HCs | 7.86±0.02 | 2266±34 | 2110±38 | 1984±38 | 98±2 | 28±2 |
| LCf period 1 | 8.13±0.02 | 2268±21 | 1992±14 | 1812±11 | 167±6 | 14±1 |
| LCf period 2 | 8.34±0.02 | 2323±23 | 1918±29 | 1662±33 | 248±8 | 8±1 |
| LCf period 3 | 7.85±0.02 | 2290±12 | 2139±12 | 2013±12 | 96±4 | 30±1 |
| HCf period 1 | 7.86±0.03 | 2332±46 | 2176±37 | 2047±33 | 100±7 | 30±1 |
| HCf period 2 | 8.10±0.02 | 2302±33 | 2042±34 | 1869±35 | 158±6 | 16±1 |
| HCf period 3 | 7.62±0.02 | 2298±6 | 2227±11 | 2115±11 | 59±2 | 53±2 |

## 2.2 Specific growth rate

Subsamples were collected and fixed with Lugol's solution for cell density measurement. Then samples were counted with a plankton counting chamber (DSJ-01, Xundeng) under an optical microscope (DM500, Leica). Specific growth rate was calculated according to the following equation: $\mu = \ln(N2/N1)/(T2-T1)$, where N1 and N2 represent cell densities at T1 and T2, respectively.

## 2.3 Chlorophyll *a* and biogenic silica contents

Subsamples were filtered onto GF/F filters (Whatman) for subsequent chlorophyll *a* extraction in 100% methanol at 4 ℃. Then they were centrifuged at 5000×g for 10 min before the absorption of supernatant was determined at 632, 665, and 750 nm with a spectrophotometer (Ultrospect 3300 pro, Amersham Bioscience). Biogenic silica (BSi) samples were collected onto polycarbonate membranes (ATTP02500, Millipore). Membranes with cells were digested in NaOH at 95 ℃ for 45 min, and then HCl was added to terminate extraction. Then ammonium molybdate and mixture of metol-sulfite, oxalic acid, sulfuric acid, and MilliQ water were added and let the color develop for 2 h. Then the absorption of the samples was determined at 810 nm to measure the BSi concentration (Brzezinski and Nelson, 1995). Cell concentration, filtration volume, and dilution factor during extraction and measurement were taken into account for calculating chlorophyll *a* and BSi contents.

## 2.4 Quantum yield of PSII

The AquaPen Chlorophyll Fluorometer (AP-C100, Photon Systems Instruments) was used to measure effective quantum yield of PSII ($\Phi_{PSII}$). Subsamples were maintained under the same light and temperature conditions as the culturing environment for at least 15 min prior to measurements. For fluorometer settings, blue light was chosen and the saturating pulse was set at 100%. It was calculated as: $\Phi_{PSII} = (F_m' - F_t) / F_m'$, where $F_m'$ and $F_t$ represent the maximum chlorophyll fluorescence of light-adapted samples and the steady state chlorophyll fluorescence, respectively.

## 2.5 Net photosynthetic and mitochondrial respiration rates

Subsamples were gently filtered (< 0.02 MPa) onto cellulose acetate membranes and then re-suspended into 20 mmol l$^{-1}$ Tris-buffered medium. The pH values of the Tris-buffered media were pre-adjusted with HCl and NaOH to the corresponding culturing values. Then re-suspended samples were injected into the chamber of a Clark-type oxygen electrode (Oxygraph plus, Hansatech) and the changes in the oxygen level were recorded for at least 10 min for each sample. Light intensity was set as 150 μmol photons m$^{-2}$ s$^{-1}$ for net photosynthetic rate measurement, and a halogen lamp (QVF135, Philips) was used as the light source. For mitochondrial respiration rate measurement, the chamber was covered by aluminum foil to achieve dark condition. Temperature of water jacket of chamber was controlled at 20 °C with a thermostatic water bath (DHX-2005, Xianou). Net photosynthetic and respiration rates were measured at each culturing pH level in fluctuating regimes. Cell density after concentrating was counted as mentioned above to calculate net photosynthetic and mitochondrial respiration rate per cell.

## 2.6 Fatty acid composition

Cells were collected by gentle filtration (< 0.02 MPa) and centrifugation (3500×g, 5 min), and then samples were dried (80 °C, 36 h) and pulverized to fine powders. Then fatty acids (FAs) in samples were converted into fatty acid methyl esters

(FAMEs) by chloroform-methanol (V:V=2:1), and their compositions were analyzed with a Shimadzu GC-2010 gas chromatograph-flame ionization detection equipped with a fused silica column (100 m × 0.25 mm × 0.2 μm film thickness, Agilent CP-Sil 88). Standards were used to identify FAMEs by comparing retention times and proportions of FAs were quantified by the percentage of each peak area to total area.

## 2.7 Statistical analysis

All data are reported as the mean ± standard deviation (SD). Shapiro-Wilk and Levene tests were used to test the normality and equal variance of data, respectively. One-way analysis of variance (ANOVA) and *post hoc* Tukey-Kramer test were used to analyze the differences among four treatments.

## 3 Results

### 3.1 Specific growth rate, chlorophyll *a* and BSi contents

There were no significant effects of pH treatments, including both mean levels and variability, on the specific growth rates of *S. costatum* and *T. weissflogii* (p = 0.103 and 0.661, respectively), although growth rates varied across sampling periods (Fig. 2). Similarly, pH treatments did not significantly affect the chlorophyll *a* content of *S. costatum*, despite lower average values under HC conditions (Fig. 3a). This lack of significance was attributed to relatively high variance in the HC treatments. For *T. weissflogii*, no differences were observed between steady and fluctuating regimes under either LC or HC conditions. HCs cells had 44% more chlorophyll *a* content compared to LCs cells (p = 0.002), but LCf and HCf cells showed similar content (Fig. 3c. p = 0.999). In terms of BSi content, both species had similar content regardless of treatments (Fig. 3b and d. p = 0.312 and 0.600 for *S. costatum* and *T. weissflogii*, respectively).

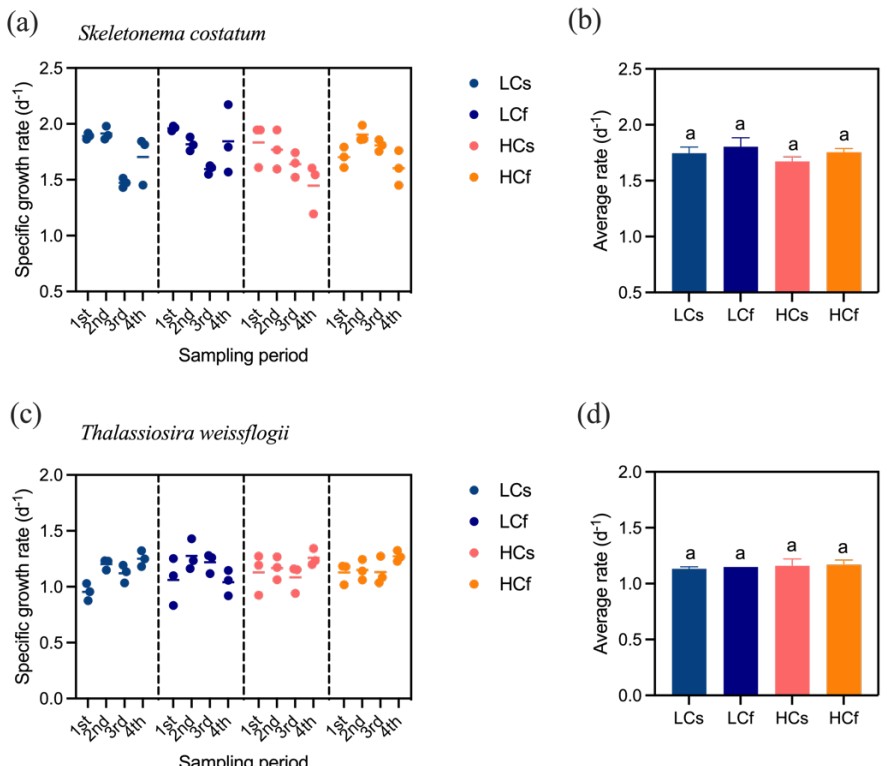

Figure 2. Specific growth rate of *S. costatum* and *T. weissflogii* cells grown under steady ambient pH/pCO$_2$ (LCs), fluctuating ambient pH/pCO$_2$ (LCf), steady future pH/pCO$_2$ (HCs), and fluctuating future pH/pCO$_2$ (HCf). Scatter plots show the specific growth rate of all replicates in each sampling period and the short line indicates the mean value of three replicates. Four sampling periods include all pH levels in the fluctuating regime and average rates are calculated from the means of triplicate cultures across four sampling periods. The different letters indicate significant ($p < 0.05$) differences among treatments.

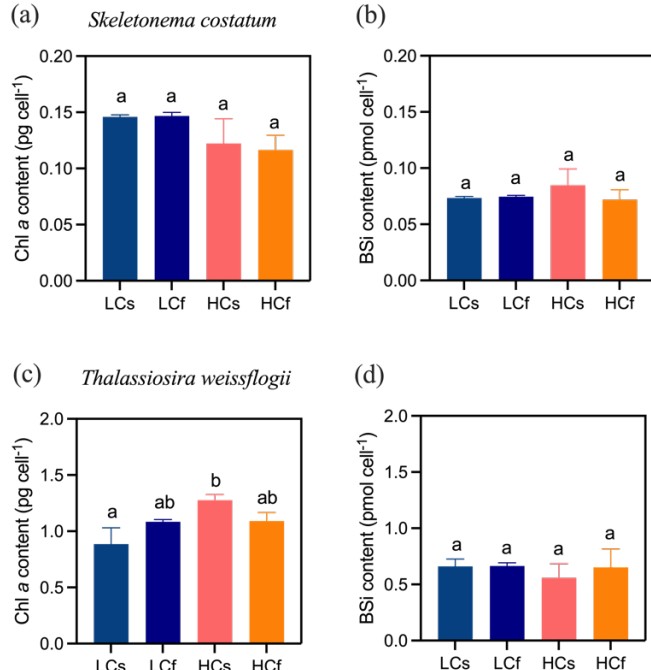

Figure 3. Chlorophyll *a* and biogenic silica contents of *S. costatum* and *T. weissflogii* cells grown under steady ambient pH/pCO$_2$ (LCs), fluctuating ambient pH/pCO$_2$ (LCf), steady future pH/pCO$_2$ (HCs), and fluctuating future pH/pCO$_2$ (HCf). Values are the means ± SD of triplicate cultures. The different letters indicate significant (p < 0.05) differences among treatments.

## 3.2 Quantum yield of PSII

Lower pH enhanced effective quantum yield of PSII of *S. costatum* by 9% and 13% compared to ambient pH condition for steady and fluctuating regimes, respectively (Fig. 4a. p = 0.001 for steady regimes and < 0.001 for fluctuating regimes). No difference between steady and fluctuating regimes was found under LC or HC conditions. Similarly, no difference between steady and fluctuating regimes was found for *T. weissflogii*, and effects of decreased pH were only observed in the fluctuating regime, with 10% higher effective quantum yield of PSII in HCf cells than LCf ones (Fig. 4b. p = 0.032).

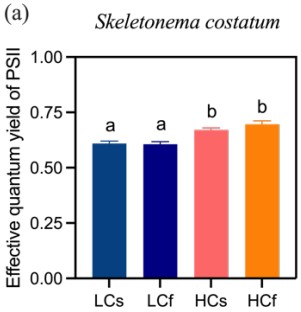 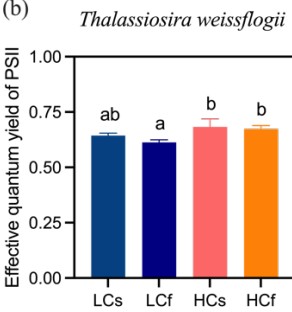

Figure 4. Effective quantum yield of PSII of *S. costatum* and *T. weissflogii* cells grown under steady ambient pH/pCO$_2$ (LCs), fluctuating ambient pH/pCO$_2$ (LCf), steady future pH/pCO$_2$ (HCs), and fluctuating future pH/pCO$_2$ (HCf). Values are the means ± SD of triplicate cultures. The different letters indicate significant ($p < 0.05$) differences among treatments.

### 3.3 Net photosynthetic and mitochondrial respiration rates

In the steady regime, seawater acidification inhibited net photosynthetic rate of *S. costatum* by 27% ($p = 0.005$), while no effects of decreased pH on average net photosynthetic rates in the fluctuating regime were observed (Fig. 5a). Although photosynthetic rate changed with pH in the fluctuating regime (Fig. 5b), the average rates of cells cultured under LCf and HCf conditions were similar with the rates under corresponding steady conditions (Fig. 5a). For *T. weissflogii*, its photosynthetic rate was generally insensitive to changes in mean level of pH or pH variability (Fig. 5e. $p = 0.345$), and no general relationship between net photosynthetic rate and pH was observed in fluctuating regimes (Fig. 5f).

Seawater acidification enhanced mitochondrial respiration rate of *S. costatum* by 36% in the steady regime ($p = 0.025$), while there was no difference in the rate between LCf and HCf conditions (Fig. 5c). *S. costatum* cells cultured under LCf condition showed increased mitochondrial respiration rate with increasing pH levels in the regime, while the rates of HCf cells were similar among three pH levels (Fig. 5d). For *T. weissflogii*, its mitochondrial respiration rate varied at different pH levels in the fluctuating regime (Fig. 5h), but the average rates of LCf and HCf conditions were similar with the rates under corresponding steady conditions, and no effects of seawater acidification were found (Fig. 5g. $p = 0.982$).

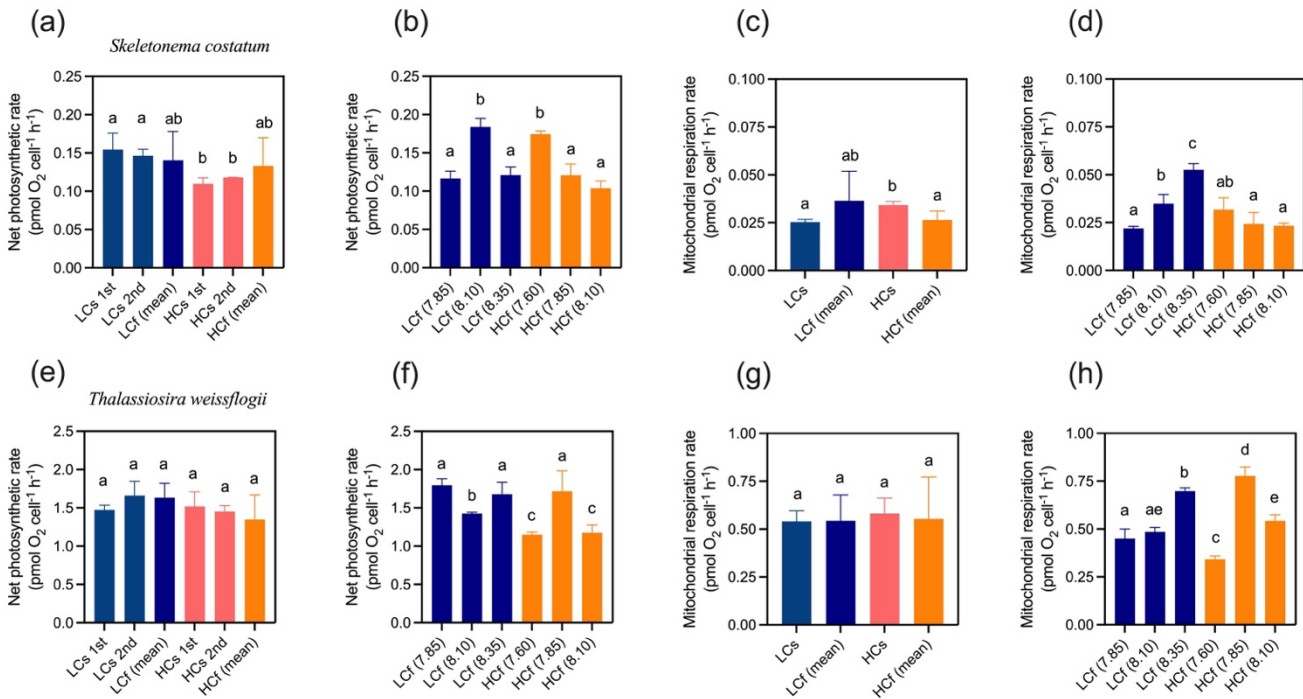

Figure 5. Net photosynthetic rate and mitochondrial respiration rate of *S. costatum* and *T. weissflogii* cells grown under steady ambient pH/pCO₂ (LCs), fluctuating ambient pH/pCO₂ (LCf), steady future pH/pCO₂ (HCs), and fluctuating future pH/pCO₂ (HCf). Values are the means ± SD of triplicate cultures. The different letters indicate significant ($p < 0.05$) differences among treatments. In the steady regime, net photosynthetic rate was measured twice on different days to monitor potential variations over time; in the fluctuating regime, net photosynthetic rate and mitochondrial respiration rate at each pH level was measured.

## 3.4 Fatty acid composition

The proportion of saturated FA (SFA) of *S. costatum* slightly decreased at lower pH, while the proportions of monounsaturated FA (MUFA) and polyunsaturated FA (PUFA) increased slightly (Fig. 6a. $p = 0.001$, 0.015, $< 0.001$ for SFA, MUFA, and PUFA, respectively). Effects of pH fluctuation were only observed for PUFA of *S. costatum* at ambient pH level, with 2% lower proportion under LCf condition ($p = 0.001$). FA compositions of *T. weissflogii* were markedly altered by seawater acidification rather than pH fluctuation, with 13% lower SFA and twofold increase in PUFA compared with ambient pH level (Fig. 6b. $p = 0.001$ and $< 0.001$ for SFA and PUFA, respectively). The decrease in SFA proportion of *T. weissflogii* was contributed by all main SFA except C17:0, which showed higher proportion under seawater acidification conditions (Fig. 7). Enhanced PUFA mainly resulted from higher eicosapentaenoic acid (EPA) and docosahexaenoic acid (DHA) under seawater acidification conditions regardless of steady or fluctuating regimes ($p < 0.001$ for EPA and DHA).

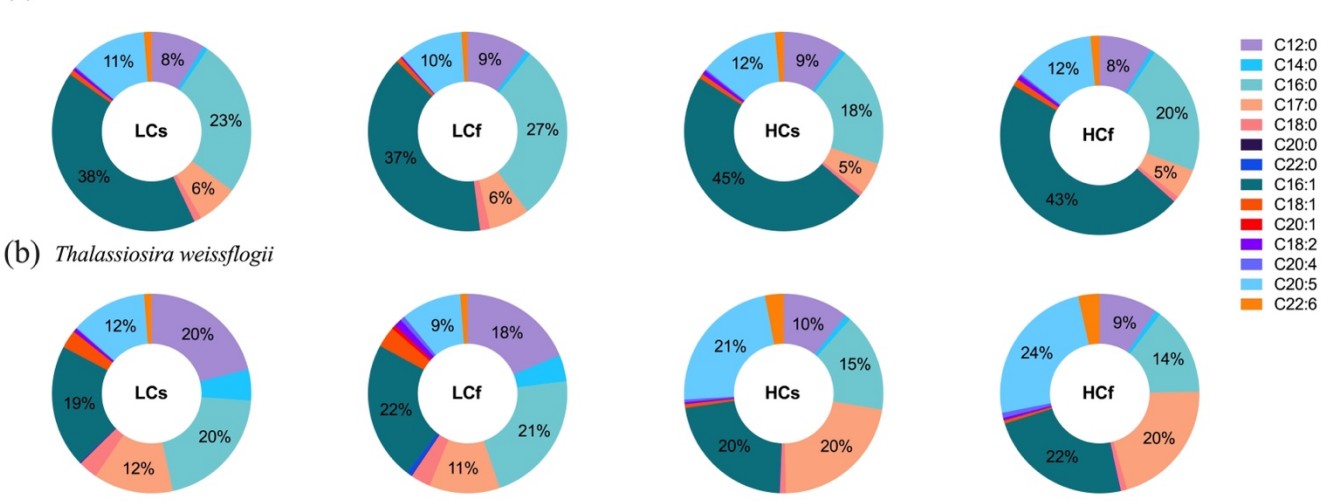

Figure 6. Saturated FA (SFA), monounsaturated FA (MUFA), and polyunsaturated FA (PUFA) proportions of *S. costatum* and *T. weissflogii* cells grown under steady ambient pH/pCO$_2$ (LCs), fluctuating ambient pH/pCO$_2$ (LCf), steady future pH/pCO$_2$ (HCs), and fluctuating future pH/pCO$_2$ (HCf). Values are the means of triplicate cultures.

Figure 7. FA composition of *S. costatum* and *T. weissflogii* cells grown under steady ambient pH/pCO$_2$ (LCs), fluctuating ambient pH/pCO$_2$ (LCf), steady future pH/pCO$_2$ (HCs), and fluctuating future pH/pCO$_2$ (HCf). Values are the means of triplicate cultures.

Table 2. Physiological responses of phytoplankton to steady and fluctuating seawater acidification. "+" indicates a positive response; " − "indicates a negative response; "ns" indicates no response.

| Phytoplankton species | Steady or Fluctuating regime | pCO$_2$ level | Duration of a entire fluctuation period | Growth | Photosynthetic rate | Chl $a$ content | BSi content | Calcification rate | Reference |
|---|---|---|---|---|---|---|---|---|---|
| *Ostreococcus* | Fluctuating | 700-1300 ppm | 7 d | + or ns | + or ns | | | | Schaum and Collins, 2014 |
| | Steady | 1000 ppm | | + or ns | + or ns | | | | |
| *Thalassiosira weissflogii* | Fluctuating | 870-1949 ppm | 24 h | ns | ns | ns | ns | | Li et al., 2016 |
| | Steady | 1005 ppm | | ns | ns | ns | ns | | |
| | Fluctuating | 400-1900 ppm | 24 h | ns | + | ns | ns | | Shang et al., 2024 |
| | Steady | 1000 ppm | | ns | ns | − | ns | | |
| | Fluctuating | 400-1750 ppm | 5 d | ns | ns | ns | ns | | Present study |
| | Steady | 1000 ppm | | ns | ns | + | ns | | |
| *Thalassiosira pseudonana* | Fluctuating | 400-1900 ppm | 24 h | ns | ns | ns | ns | | Shang et al., 2024 |
| | Steady | 1000 ppm | | ns | ns | ns | ns | | |
| *Thalassiosira oceanica* | Fluctuating | 870-1949 ppm | 24 h | − | ns | ns | − | | Li et al., 2016 |
| | Steady | 1005 ppm | | − | − | + | ns | | |
| *Skeletonema costatum* | Fluctuating | 400-1750 ppm | 5 d | ns | ns | ns | ns | | Present study |
| | Steady | 1000 ppm | | ns | − | ns | ns | | |
| *Emiliania huxleyi* | Fluctuating | 600-1800 ppm | 24 h | + | ns | ns | | − | Li et al., 2021 |
| | Steady | 1200 ppm | | ns | ns | ns | | − | |

## 4 Discussion

### 4.1 Effects of pH changes on the physiological performance of two diatoms

Previous studies have shown that *T. weissflogii* and *S. costatum* exhibit high tolerance to changes in pH levels. *T. weissflogii* has been found to tolerate decreased pH/increased $pCO_2$, while seawater alkalization significantly inhibited growth when the medium pH exceeded 8.44 (Li et al., 2019). For *S. costatum*, growth remained nearly constant within a pH range of 6.5 to 8.5 (Taraldsvik and Myklestad, 2000), and the effective quantum yield of PSII in cells acclimated to the seawater acidification condition was unresponsive to pH changes between 7.6 and 8.2 (Zheng et al., 2015). In the present study, growth remained unchanged in both diatoms, although chlorophyll content, net photosynthetic rate, and the quantum yield of PSII were either enhanced or inhibited, depending on the species. Growth, as a proxy of the overall cellular response, is influenced by a range of metabolic processes and may remain insensitive to pH variations, despite changes in photosynthetic parameters.

The DBL exists at the interface of cells, in which the microenvironment is different from the bulk seawater. For example, pH in DBL could increase substantially when photosynthesis happens inside the cell, and larger cells could experience more significant changes in pH within DBL (Chrachri et al., 2018). Thus, just like the condition in coastal waters, stable seawater carbonate chemistry is not realistic in DBL of large cells. For this reason, it seems reasonable to expect neutral or positive effects of pH fluctuation on coastal phytoplankton. Our previous comparative-study showed that diurnal pH fluctuations depressed the growth and photosynthesis of the oceanic diatom *Thalassiosira oceanica* under either ambient or low pH conditions, whilst coastal diatom *T. weissflogii* was insensitive to pH fluctuations, and enhanced production rate of particulate organic carbon was even observed in fluctuating regimes (Li et al., 2016). In that study, seawater carbonate chemistry was manipulated to achieve high-frequency pH change, with a full pH fluctuation period of 24 hours (pH increased during the light period and decreased during the dark period, with the $CO_2$ partial pressure of the aerating gas adjusted every 12 hours). This approach provided a holistic view of phytoplankton response to fluctuating pH, but lacked specific analysis of the response to each pH period within the fluctuating regime. The high pH/low $CO_2$ period might have negative effects if the treatment duration extends beyond several hours. In the present study, the fluctuation period was set to 5 days, and we found that the average net photosynthetic and respiration rates of *T. weissflogii* were unaltered by treatments (Fig. 5e), although these rates varied at different pH levels in fluctuating regimes (Fig. 5f). For *S. costatum*, no differences were observed in average photosynthetic rates between steady and fluctuating regimes under ambient and low pH levels. However, cells cultured under steady acidification conditions exhibited a 27% reduction in net photosynthetic rate and a 36% increase in respiration rate compared to cells grown under steady ambient pH.

The two species in this study exhibited different photosynthetic responses to seawater acidification, potentially due to their distinct inorganic carbon utilization strategies. The biochemical $CO_2$ concentrating mechanism (i.e. unicellular $C_4$ pathway) has been suggested to play a key role in the plasticity of inorganic carbon utilization in *T. weissflogii* (Reinfelder et al., 2000). This mechanism is crucial for maintaining relatively high photosynthetic rates in responses to changes in pH. Additionally,

differences in membrane permeability and fluidity may contribute to these species' varied responses. FAs are incorporated into phospholipids, which are major structural components of cell membranes, and FA composition can influence membrane characteristics. Theoretically, the cell membranes of *T. weissflogii* are less fluid and less permeable to $CO_2$ (Maulucci et al., 2016), due to higher proportions of SFA (Fig. 6). This helps cells cope with decreased intracellular pH and maintain homeostasis under seawater acidification conditions (Rossoll et al., 2012).

Although there was tenfold difference in net photosynthetic rates between the two species, their chlorophyll *a*-normalized photosynthetic rates were comparable. This is because the larger *T. weissflogii* contains ten times more chlorophyll *a* than *S. costatum*. Seawater acidification increased chlorophyll *a* content in *T. weissflogii*, while the content tended to decrease in *S. costatum*. The "pigment economy" hypothesis, proposed in previous studies, explains the reduction in pigment content under seawater acidification conditions by suggesting that cells eliminate chlorophyll molecules that are inefficient for light capture during photosynthesis (Gordillo et al., 1998, 2015). However, this hypothesis does not apply to *T. weissflogii*, indicating that different species employ distinct strategies for pigment synthesis in response to acidification. Both species in this study maintained stable BSi content regardless of pH changes. In contrast, the content and production rate of BSi in the oceanic diatom *T. oceanica* decreased under seawater acidification and fluctuating pH conditions (Li et al., 2016). Researce on the effects of fluctuating pH on oceanic phytoplankton is limited, and further studies are needed to determine whether oceanic species are generally more susceptible to fluctuating pH than coastal species.

Based on the results of the present and previous studies, the effects of seawater acidification may differ depending on whether pH fluctuations are considered in experiments (Table 2). While the trends between steady and fluctuating seawater acidification regimes are similar, the variation amplitudes differ. If the steady regime was used to simulate coastal acidification, inhibited primary production could be observed in some species. However, no such effects were observed when coastal acidification was simulated under more realistic fluctuating pH conditions. In the complex and turbulent environment of coastal waters, dynamic pH fluctuations may have more diverse influences on phytoplankton than steady pH. Therefore, it is important to consider pH variation for more accurate predictions regarding the consequences of acidification in coastal waters.

## 4.2 Fatty acid composition was altered by pH mean level

Omega-3 long-chain essential fatty acids are integral to key functions in aquatic and terrestrial organisms. These FAs are directly or indirectly contributed by phytoplankton in marine ecosystem (Hixson and Arts, 2016). Among them, DHA and EPA are well known due to their benefits in enhancing nutritional quality of marine primary consumers (Kainz et al., 2004), especially herbivorous copepods and rotifers, which serve as prey for secondary consumers such as fishes and crustaceans. The only way of obtaining essential FAs for marine animals is through their diet, as they cannot synthesize them *de novo* (Brett and Müller-Navarra, 1997). EPA plays a critical role in growth, development, and reproduction of marine consumers, and diatoms are the main producers of EPA in marine ecosystem (Budge et al., 2014). EPA is the major PUFA in both species tested here, with its proportion accounting for more than 10% in total FA under all conditions. DHA is the second

most abundant PUFA in two diatoms, although its proportion is much lower than EPA. The proportions of EPA and DHA observed in this study fall within the ranges reported for marine microalgae (Boelen et al., 2013). However, it is important to note that these proportions can vary significantly both between species and within a single species under different conditions. For example, the growth phase of cells can have a substantial impact on fatty acid composition (Schwenk et al., 2013). Therefore, it is crucial to specify the growth phase of cells when comparing results with other studies. In the present study, cells were maintained in the exponential phase under conditions with sufficient nutrients. These conditions are not optimal for lipid accumulation, suggesting that there is potential for further enhancement of essential FAs.

Species- or even strain-specific responses of FAs to ocean acidification have been documented in previous studies. In terms of FA composition, different phytoplankton strains of one species appeared to respond to increased $CO_2$ in a varied way. For instance, the EPA and DHA fractions of total FA in a highly $CO_2$-tolerant strain of *T. weissflogii* were lower when cells were cultured under 5%, 10%, and 20% $CO_2$, compared with control air condition (Ishida et al., 2000). While EPA and DHA or total PUFA proportions of *T. weissflogii* (CCMP2599) were not altered when $pCO_2$ levels increased from 320 to 690 and 2900 ppm (King et al., 2015). These studies indicate that changes in FAs in response to seawater acidification are species- or strain-dependent and influenced by the culture conditions employed in studies. Although the mechanisms underlying these changes remain unclear, they are believed to be related to the regulation of lipid metabolites (Jin et al., 2021) and processes closely associated with lipid metabolisms, such as carbon concentration mechanisms (Abreu et al., 2020) and the regulation of intracellular pH homeostasis (Rossoll et al., 2012). It has been proposed that pH may act as a regulatory signal for the formation of cell membranes by controlling the production of the synthesizing enzymes of FAs (Young et al., 2010).

In the present study, PUFA and MUFA of *S. costatum* were slightly enhanced by decreased pH, whilst PUFA of *T. weissflogii* increased substantially, without changes observed in MUFA. The increased proportion of PUFA was mainly attributable to higher EPA and DHA proportions. Given the varied responses among species, the effects of ocean acidification on FA composition of phytoplankton community would be mediated by community structure. Indeed, the increase in PUFA in response to seawater acidification is likely linked to changes in taxonomic composition, as reported in a study of a natural plankton community in the Arctic (Leu et al., 2013). These changes could be transferred to higher trophic levels through the marine food web. This is evidenced by the study in a mixed phytoplankton assemblage including *T. weissflogii*, which found that seawater acidification was conducive to the accumulation of unsaturated FAs (Wang et al., 2017). However, for copepods fed a high-$pCO_2$ *Thalassiosira pseudonana* diet, a decrease in both copepod somatic growth and egg production was found, attributed to the lower PUFA content in their diet (Rossoll et al., 2012). Thus, seawater acidification might have complex and far-reaching consequences for marine ecosystem functioning through altering intracellular macromolecules in primary producers.

## 5 Conclusions

Although temperature, light intensity, and nutrient limitation usually have prominent influences on photosynthetic performance and nutritional quality of primary producers, the regulating effects of seawater acidification should not be ignored, especially given the cascading effects throughout marine food webs. In the present study, the growth and BSi content of both species, and photosynthetic and respiration rates of *T. weissflogii*, were impacted by neither decreased pH nor pH fluctuation, indicating their tolerance to pH changes. Nevertheless, fatty acid compositions of two species were altered by seawater acidification, with lower SFA and higher PUFA proportions compared to ambient pH condition. Although the deterioration of nutritional quality (Rossoll et al., 2012) and lower production of PUFA (Hixson and Arts, 2016) were projected in the more warmed and acidified ocean, our results suggest that seawater acidification may enhance PUFA production without affecting growth in coastal diatoms, particularly *T. weissflogii*. Furthermore, taking dynamic carbonate chemistry into account would help investigate and predict consequences of coastal acidification more properly.

*Data availability.* Data presented in this study have been deposited in the Zenodo repository: https://doi.org/10.5281/zenodo.13142180 (Shang et al., 2024).

*Author contributions.* YS, JQ: methodology, formal Analysis, investigation, writing-original draft; YW, XW, YZ: investigation; JX, DZ: conceptualization, writing-review & editing, supervision; FL: conceptualization, methodology, validation, investigation, writing-review & editing, funding acquisition.

*Competing interests.* The authors declare that they have no conflict of interest.

*Financial support.* This research has been supported by the National Natural Science Foundation of China (No. 42206138), the Natural Science Foundation of Jiangsu Province (No. BK20220684), the "Qinglan Project" of Jiangsu Province, the "521 High-Level Talent Training Project" of Lianyungang, the Innovation and Entrepreneurship Training Program of Jiangsu Province, and the Priority Academic Program Development of Jiangsu Higher Education Institutions.

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
