# Peer review of "Effects of pH/pCO2 fluctuations on photosynthesis and fatty acid composition of two marine diatoms, with reference to consequence of coastal acidification"

_EGUsphere, 2024_

## Author Response (AR1)

RC1

The study examined the effects of steady and fluctuating pH/pCO2 conditions on two coastal diatom species, Skeletonema costatum and Thalasiosira weissflogii. The authors found that seawater acidification had neutral or positive impacts on growth, chlorophyll, and silica content. However, elevated pCO2 inhibited photosynthesis and enhanced respiration in S. costatum under steady pH, but not under fluctuating pH. The results suggest coastal acidification can affect primary production and nutritional quality differently from open ocean acidification, with potential consequences for marine ecosystem functioning. The major highlight of this study is it examines the effect of fluctuating pH or pCO2 on the diatom physiology instead of stable pH/pCO2. This aligns better with the situations in the marine environment, which is very helpful to predict the outcome of seawater acidification in the future.

Response: We appreciate your recognition of the significance of our work and thank you for the through and professional review as well as the valuable feedback on our manuscript. Most published studies have investigated the effects of ocean acidification under stable pH conditions. However, the carbonate chemistry parameters of coastal seawaters are variable, particularly on shorter time scales. Therefore, it is important to consider pH fluctuations in ocean acidification studies.

I have a few comments on the manuscript:

Title: Should be fluctuations

Response: Thank you for the suggestion. We have incorporated this in the revised manuscript.

Line 15-18: Elevated pCO2 leaded to 21% lower saturated… As this study evaluate both stable and fluctuating pCO2 scenarios. The authors need to clearly explain which scenario they are referring to. The statement in the abstract seems different from the conclusion that "In the present study, the growth and most parameters of two typical coastal diatom were impacted by neither decreased pH nor pH fluctuation…". Please check.

Response: As suggested, we have provided additional information here to clarify and avoid potential confusion. The revised version now reads as "Acidification conditions lead to lower saturated FA and higher polyunsaturated FA proportions in both species, regardless of steady and fluctuating regimes" (Line 17-18)

In the conclusion, we have specified the particular parameters that were unaffected, rather than using the term "most parameters". (Line 304-306)

Line 25: wind forcing—wind

Response: We have made modifications as suggested.

 Line 24: carbonate system parameters—seawater chemistry

Response: We have revised "carbonate system parameters" to "seawater carbonate chemistry" as suggested.

Line 26: This is a clause, so the period should be changed to a comma. Additionally, the authors have used too many restrictive clauses in the writing process. Too many "which" in the text. "in which they contribute", "which is supported", "in which cells show"…

Response: Thank you for highlighting this point. We agree that using too many restrictive clauses is not ideal for the manuscript. In the revised version, we have modified these sentences to reduce the use of "which" where possible.

Line 29: Not seem to be, they are.

Response: Agree. We have removed "seem to be".

Line 36: amplitude and high frequency of what?

Response: Our intention was to describe that the amplitude and frequency of pH/pCO2 variations in coastal waters are substantial and occur at a high rate. We have rephrased this sentence to clarify. The revised version now reads as: "In addition, coastal waters are characterized by large amplitude and high-frequency fluctuations in seawater carbonate chemistry parameters, particularly pH and $pCO_2$". (Line 38-39)

Line 40: show—have

Response: We have made this modification as suggested.

Line 42: should emphasize "stable" in the sentence. The impacts of fluctuating carbonate chemistry may differ from the effects of a stable decrease in pH and increase in CO2.

Response: Agree. We have added "in steady regimes" to this sentence to enhance clarity.

Line 44: These studies hinder our understanding of impacts of acidification in coastal regions?

Response: We have revised the sentence to "Limited studies have focused on how marine phytoplankton perform under fluctuating pH/$pCO_2$ condition, leaving the effects of coastal acidification poorly understood. This knowledge gap impedes accurate predictions of acidification impacts in coastal regions." (Line 48-50)

Line 52-55: Were the diatoms acclimated before experiments?

Response: Two diatom species were acclimated to four treatments for a minimum of 10 days before measurements were taken. Prior to being subjected to the four pH/pCO2 conditions, the cells were maintained under steady ambient and projected future pH/ pCO2 conditions for 7 days.

Line 54: artificial seawater. What is the ingredient of the artificial seawater?

Response: The artificial seawater contains 10 basic salts: the anhydrous salts (i.e., NaCl, Na2SO4, KCl, NaHCO3, KBr, H3BO3, NaF) and hydrous salts (i.e., MgCl2 · 6H2O, CaCl2 · H2O, and SrCl2 · 6H2O). In the revised manuscript, we have included a reference here (Sunda et al., Trace metal ion buffers and their use in culture studies, in *Algal culturing techniques* 4 (2005): 35-63.).

Line 58: diluted every three or four days. What were the cell densities in this period? These diatoms are fast-growing species.

Response: Cells were inoculated at very low initial cell density (fewer than 500 cells/ml for *S. costatum* and fewer than 50 cells/ml for *T. weissflogii*), with maximum densities during culturing controlled bellow 160,000 cells/ml for *S. costatum* and 10,000 cells/ml for *T. weissflogii*. We have included this cell densities information in the revised manuscript (Line 62-64).

Line 60-69: It would be better to clearly explain the rationale behind the choice of pCO2 levels in either the introduction or methods section, such as the amplitude and frequency of the fluctuations, and how they compare to real-world conditions.

Response: Agree. The future pCO2 level was set according to the IPCC report, with fluctuating amplitude (±0.25 pH units) and frequency (one pH variation cycle lasting 5 days) based on habitat conditions. This information has been added. However, since seawater carbonate chemistry in coastal waters is influenced by multiple factors and pH fluctuation frequency varies across habitats, it is not feasible to cover all possible conditions in a single study. We selected the amplitude and frequency based on both the real-world conditions and experiment feasibility. (Line 69-71 and 74-75)

Fig.1 This figure only indicates the Target pH of ambient pH/pCO2. What was the measured pH in the culturing medium?

Response: Indeed, it is better to present the measured pH values here. We have made this modification in the revised MS.

Line 106: Better to convert RPM into g

Response: Thank you for pointing this out. It is indeed preferable to report centrifugal force rather than RPM, as centrifugal radii vary across different centrifuges.

Line 25: If the species name has been mentioned earlier in the text, the abbreviated name can be used subsequently.

Response: We use the abbreviated name in the revised MS.

Fig. 7: The treatments are not labeled in the figure.

Response: The treatments are actually labeled in the center of each pie chart. We have increased the font size to improve clarity.

Line 165-172: If statistics is not marked in the figure 6 and 7. It would be better to state in the text.

Response: We have included the statistical information for the results section.

Line 186-190: This paragraph appears redundant or repetitive with information presented earlier in the introduction.

Response: We have deleted this paragraph and relocated some of its sentences to the corresponding sections of the introduction and discussion.

Line 191-200: S. costatum is totally ignored here. Any comment for the response of S. costatum to the fluctuating pCO2?

Response: This section briefly summarizes findings from our previous comparative study, which focused on *T. weissflogii* and *T. oceanica*. To our knowledge, no studies have examed the effects of fluctuating pH on *S. costatum*. However, we add further discussion on *S. costatum* as suggested. The revised version now reads as: "Previous studies have shown that *T. weissflogii* and *S. costatum* exhibit high tolerance to changes in pH levels. *T. weissflogii* has been found to tolerate decreased pH/increased $pCO_2$, while seawater alkalization significantly inhibited growth when the medium pH exceeded 8.44 (Li et al., 2019). For *S. costatum*, growth remained nearly constant within a pH range of 6.5 to 8.5 (Taraldsvik and Myklestad, 2000), and the effective quantum yield of PSII in cells acclimated to the seawater acidification condition was unresponsive to pH changes between 7.6 and 8.2 (Zheng et al., 2015)." (Line 207-211)

Line 201: What is the specific frequency and how it is compared with the result in this study?

Response: In our previous study, we investigated the effects of diurnal pH fluctuations. These fluctuating regimes were achieved by adjusting the CO2 partial pressure every 12 h, resulting in continuous changes in pH. We found that the fluctuating carbonate chemistry regime had either positive or negligible effects on the physiological performance of *T. weissflogii*. We have included this information in the revised MS. "In that study, seawater carbonate chemistry was manipulated to achieve high-frequency pH change, with a full pH fluctuation

period of 24 hours (pH increased during the light period and decreased during the dark period, with the $CO_2$ partial pressure of the aerating gas adjusted every 12 hours). This approach provided a holistic view of phytoplankton response to fluctuating pH, but lacked specific analysis of the response to each pH period within the fluctuating regime." (Line 222-226)

Line 225: Kinds of effect—Various effect

Response: Thank you for the suggestion. We have made this modification.

Line 230-235: The authors only focus on the consequences of the altered FA content. But what could be the mechanisms behind the change?

Response: It has been proposed that pH might act as a regulation signal for the formation of cell membranes, which are mainly composed of fatty acids, by controlling the production of its synthesizing enzymes. We have added some discussions on the possible mechanisms involved in this process. "Although the mechanisms underlying these changes remain unclear, they are believed to be related to the regulation of lipid metabolites (Jin et al., 2021) and processes closely associated with lipid metabolisms, such as carbon concentration mechanisms (Abreu et al., 2020) and the regulation of intracellular pH homeostasis (Rossoll et al., 2012). It has been proposed that pH may act as a regulatory signal for the formation of cell membranes by controlling the production of the synthesizing enzymes of FAs (Young et al., 2010)." (Line 283-288)

RC2

**General Comments:**

This study aims to understand the physiological impact of fluctuating CO2/pH on coastal diatoms. The presented findings showed very little difference between fluctuating and stable regimes with the same average pH. This is an interesting experimental design and dataset, but needs additional editing to clarify the experimental data and more thoroughly discuss the data presented.

Response: We sincerely appreciate the constructive comments and suggestions. Thank you for your professional review of our manuscript. Your comments and suggestions are important in improving the quality of our manuscript.

Generally well written. Needs some clarification about the sampling times for each parameter over the course of the experiment. My main concern is the content of the discussion. I believe there is more to be discussed. Only 1 of 3 paragraphs in the 4.1 section directly discusses the data presented in this paper. A discussion section can give contextual information and describe facets of the results, but I would like to see more about how these specific data relate

to previous works and/or discussion of the nuances of the dataset.

Response: Thank you for highlighting this deficiency in our manuscript. We recognize that this section lacks a through and detailed discussion, particularly regarding the similarities and differences with published studies. We have expanded our discussion on the effects of stable and fluctuating seawater acidification on growth, photosynthesis, and fatty acid composition in marine diatoms, with special emphasis on the species examined in the present study.

For examples, how do the culture conditions relate to changes in photosynthesis and FA, as the growth rate and growth phase can have significant impacts of fatty acid composition. Eg. early and late exponential phase can be quite different in terms of physiology.

Response: We agree that results can vary significantly depending on the phase in which samples are obtained, even when cells are in the exponential phase. This is why we sampled four times for growth rate measurement. In the present study, cultures were diluted every three or four days, and the initial cell density was kept relatively low to ensure stable cell conditions and to prevent inhibition due to nutrient limitation. Specifically, the initial cell densities are fewer than 500 cells/ml for *S. costatum* and fewer than 50 cells/ml for *T. weissflogii*, with maximum densities during culturing controlled bellow 160,000 cells/ml for *S. costatum* and 10,000 cells/ml for *T. weissflogii*. We have included the cell density information in the revised MS (Line 62-64). Additionally, it is necessary to discuss the impacts of growth phase on cell physiology, as this is crucial for proper interpretation of results. Therefore, we have incorporated related discussions as suggested "However, it is important to note that these proportions can vary significantly both between species and within a single species under different conditions. For example, the growth phase of cells can have a substantial impact on fatty acid composition (Schwenk et al., 2013). Therefore, it is crucial to specify the growth phase of cells when comparing results with other studies. In the present study, cells were maintained in the exponential phase under conditions with sufficient nutrients. These conditions are not optimal for lipid accumulation, suggesting that there is potential for further enhancement of essential FAs." (Line 271-276)

There is no discussion of the chlorophyll data, quantum yield data, and bSi. I believe it should been mentioned, even if they don't appear to change. It would have been nice to see a non-coastal diatom for comparison showing higher susceptibility to fluctuating pCO2.

Response: Thank you for the valuable suggestion! In our previous study (Li et al., 2016. Physiological responses of coastal and oceanic diatoms to diurnal fluctuations in seawater carbonate chemistry under two CO2 concentrations, Biogeosciences), we compared the responses of the coastal diatom *T. weissflogii* and the oceanic diatom *T. oceanica*. We follow your suggestion to compare the two coastal species in the present study with the oceanic species from our previous work. We have included pigment, BSi content, and quantum yield

of PSII in these comparations (Line 249-252).

Additionally, we add a table to summarize effects of steady and fluctuating acidification on phytoplankton and related discussions. (Table1 and Line 253-260).

It was pointed out that fluctuating CO2/pH experiments are rare, but this unique design was not clearly presented in the context of the discussion. Does this prove that studies testing only conditions of stable low pH are an acceptable experimental design simplification give that the mean HCf and HCs are the same for most parameters?

Response: This applies to the responses of certain parameters in the two tested species over short-term scales (from several to tens of generations). However, our previous studies have shown that fluctuating pH can have positive or negative impacts on the physiology of the oceanic diatom *T. oceanica* (Li et al., 2016, Biogeosciences) and the coccolithophore *Emiliania huxleyi* (Li et al., 2021. Diurnally fluctuating pCO2 enhances growth of a coastal strain of *Emiliania huxleyi* under future-projected ocean acidification conditions, ICES Journal of Marine Science). In addition, long-term adaptation, species-specific response, and interactions among multiple drivers complicate these effects. However, it is impractical to test every species under all possible conditions. In this case, key species in critical ecosystems should be prioritized for investigation. We have included this information in the revised manuscript (Table 1 and Line 253-260).

Was net photosynthesis the only parameter measured at each pH level in the fluctuating time course? What about everything else? Is it presented this way because there were no differences at the fluctuating pH levels?

Response: Three important parameters-growth rate, photosynthesis and respiration rates-were measured at each pH level in the fluctuating regime. Other parameters, such as pigment, BSi, and fatty acid compositions, were only measured at the pH levels corresponding to stable conditions. The reason behind this choice is that biochemical compositions tend to be less sensitive to changes in pH within a short time compared to photosynthesis and respiration. In the present study, each pH step lasts only 24 h. Therefore, we focused on measuring photosynthesis, respiration, and growth at each pH level. The sampling information has been added in the revised MS. (Line 79-84)

**Specific Comments:**

Line 35: Are you saying that the larger-sized coastal cells that tolerate large differences between DBL and bulk seawater will be better at tolerating large swings in pH/CO2?

Response: Exactly, that is what we intended. We have rephrased this sentence for clarity. The

revised version now reads as: "This phenomenon is suggested to be related to the larger pH differences experienced by larger-sized coastal phytoplankton cells between the diffusion boundary layer (DBL) surrounding the cells and the bulk seawater (Flynn et al., 2012)." (Line 36-38)

Line 52: Are these simply two examples of coastal diatoms, or do these species have a specific differences that you intended to compare?

Response: These two species are widely used to in diatom studies and exhibit different cell sizes and inorganic carbon utilization characteristics. However, since they belong to different genera, it is inappropriate to compare their responses directly, as we cannot determine whether the differences are due to cell size or species-specificity. Nevertheless, we discuss their different responses in the discussion section.

Line 69: Can you describe the sampling frequency here? I think you sample within each 24hr period, but I am not sure.

Response: Thank you for the suggestion. For growth rate, photosynthesis and respiration rates, sampling was conducted at each pH level at 24 h intervals. We have included this information about the sampling frequency in the manuscript. (Line 79-84)

Line 70: This figure is helpful!

Response: Thank you for your encouragement! We also include the measured pH values in this figure, as suggested by another referee.

Line 90: Each sample was light-adapted before measurement?

Response: Exactly. All subsamples for effective quantum yield measurement were maintained under the same light and temperature conditions prior to measurement. This is crucial because effective quantum yield is highly sensitive to light levels, and variations in light intensity can lead to significant changes in this parameter.

Line 95: How often did you measure this? Once a day?

Response: As mentioned above, we measured photosynthetic oxygen evolution and mitochondrial respiration rates at each pH level at 24 h intervals in the fluctuating regime. This information have been included in the methods section.

Line 104: Is this Net photosynthesis? Use the same terminology in methods section that you will use in the figures.

Response: Yes, the "oxygen evolution rate" is equivalent to net photosynthesis. As suggested, we use consistent terminology in the revised manuscript.

Line 106: Are FA samples only from the end of the experiment?

Response: Yes, FA measurements are conducted only at the end of the experiment because they require high biomass. We controlled the cell density in cultures to ensure the cells remained in the exponential phase. Thus, large volumes are needed to meet the requirement for FA measurement.

Line 118: I am not sure what you mean by "only pH" is shown , as you have figures that are labeled by your pH/Co2 treatments (LCs, HCs, etc). Please clarify. Can you just say "treatment" instead of using "pH level"?

Response: This suggestion is helpful. We have made this modification as suggested. The revised version now reads as: "There were no significant effects of pH treatments, including both mean levels and variability, on the specific growth rates of *S. costatum* and *T. weissflogii* (p = 0.103 and 0.661, respectively), although growth rates varied across sampling periods (Fig. 2)." (Line 133-135)

Fig 2: The growth rates between sampling periods decline in Skeletonema cultures? Were these cultures reaching stationary phase/late exponential during your study length?

Response: Cultures were diluted every three or four days to maintain the cell density below 160,000 cells/ml for *S. costatum*. Based on our previous results from batch cultures of *S. costatum*, the maximum cell density can reach ca. 1,200,000 cells/ml (Qiu et al., 2022. Comparative study of the physiological responses of *Skeletonema costatum* and *Thalassiosira weissflogii* to initial pCO2 in batch cultures, with special reference to bloom dynamics, Marine Environmental Research). Therefore, the cells are in the middle of exponential phase in the present study. We think that the observed variation in growth rate is not caused by the stationary phase or late exponential phase. Decreasing or partially decreasing trends are noted in both LCs and LCf cultures, suggesting that the effect of fluctuating pH can be excluded. Additionally, variations in growth rates within a single treatment, despite inconsistent trends, are also observed in *T. weissflogii*. Thus, growth rates should be measured multiple times to calculate an average rate, rather than relying on a single measurement, even in steady treatments.

Line 120: I know Skeletonema Chl content may not be statistically different between LC and HC (Fig.3), but is it worth mentioning the higher variance in the HC treatments and visually lower averages?

Response: Some previous studies have observed changes in chlorophyll content under HC conditions. One hypothesis suggests that the decrease in pigment content at high CO2 relates to the elimination of chlorophyll molecules that are in excess and would not participate efficiently in the light capture for photosynthesis (Gordillo et al., 2015. Ocean acidification modulates the response of two Arctic kelps to ultraviolet radiation, Journal of Plant Physiology). This is related the "pigment economy" hypothesis (Gordillo et al., 1999. Effects

of increased atmospheric CO2 and N supply on photosynthesis, growth and cell composition of the cyanobacterium *Spirulina platensis* (Arthrospira), Journal of Applied Phycology). We have included related discussions on the "pigment economy" in the revised version. (Line 242-248)

Figure 5: Can you make this a 4 panel figure?? I want to know if the letters indicate statistical difference between LCf and LCs, but I cant tell from this setup. I THINK you have each panel stats indicated individually, but in the discussion you compare LC v HC, so it would be nice to have those in the same panel.

Response: Thank you for your suggestion. We integrate the LC and HC panels of each species into a single panel. In addition, we add letters above columns to facilitate comparisons between LCs and LCf ones, as well as LC and HC conditions. Please see the new version of Figure 5.

In Figure 5 you present fluctuating regime data where you measured the culture every 24hr period. Do you have this same data (daily data) for the stable regime? It would be useful to see what kind of variability exists within a culture across the growth curve. For example, growth rate seems to decrease more over the 4 sampling periods for S. costatum

Response: For the stable regime, we don't have daily data; however, we measured photosynthetic rates twice. Samples for growth and photosynthesis measurements were obtained on the same sampling day. Given the variations in growth rate, it is reasonable to include data from both measurements in the figure,. We add the rates measured at the other sampling point to the figure.

Line 165: Hard to assess this assertion without stats.

Response: Another referee also points this out. We have included the statistical information in the results section.

Line 167: Which figure? Fig 6a. I don't see a 14% decrease in S. costatum

Response: The amplitude of variation (14%) in this context is ambiguous. We intended to indicate that the percentage of PUFA in the LCf treatment (12%) was lower than that in the LCs treatment (14%). The decrease in percentage is 2%, which gives an amplitude of variation relative to LCs of 2%/14%=14%. However, this is indeed confusing. The revised version now reads as: "Effects of pH fluctuation were only observed for PUFA of *S. costatum* at ambient pH level, with 2% lower proportion under LCf condition ($p = 0.001$)." (Line 186-187)

Line 203: You say here that T. weissflogi is unaltered in any Ph level in the fluctuating regime, but Fig 5 e shows pH 7.85 and pH 8.1 as statistically different…

Response: Indeed, the photosynthetic rate at pH 7.85 is higher than that at pH 8.1. The revised version now reads as: "In the present study, the fluctuation period was set to 5 days, and we found that the average net photosynthetic and respiration rates of *T. weissflogii* were unaltered by treatments (Fig. 5e), although these rates varied at different pH levels in fluctuating regimes (Fig. 5f)." (Line 227-229)

Thank you for pointing thin out.

Also, are 7.6 and 7.85 truly not statistically different in panel f?? That is visually surprising

Response: We are also surprised by this and we now understand the reason. When we compare the rates at pH 7.6 and pH 7.85 directly using an independent t-test, the difference is significant. However, the independent t-test is not suitable for comparing multiple groups. Tukey is the most commonly used post hoc method and is recommended when there are equal sample sizes. In our analysis, we pooled data from three pH levels to get the "HCf mean", resulting in a sample size of 9, while the other groups have a sample size of 3. When we calculate the average rate at each pH level and then combine these three average values (sample size of 3), the difference between pH 7.6 and pH 7.85 becomes significant. Thus, the unequal sample sizes biased the post hoc results. We now use the three average values for the "HCf mean" and "LCf mean" instead of simply pooling the data together, and a *post hoc* Tukey-Kramer test is applied to provide statistical information consistent with visual observations.

Line 205: I'm not sure what you are trying to say here. Do you want to point out that the HCf cuktures do not have the same maximum Net rate at 8.1? Why do you think that happens?

Response: We intended to illustrate that the trends of photosynthesis vs. pH differ between LCf and HCf conditions. However, we now think it is inappropriate to discuss the trend with only three pH points. Our current data can only demonstrate that photosynthetic rates vary at three pH levels in the fluctuating regime. Thus, we focus on this observation rather than on the photosynthesis vs. pH relationship. We have rephrased this sentence accordingly "For *S. costatum*, no differences were observed in average photosynthetic rates between steady and fluctuating regimes under ambient and low pH levels." (Line 229-230)

These are both coastal diatoms, why do they respond to pH levels in the fluctuating regime so differently?

Response: This phenomenon is worth mentioning and discussing here. The revised version now reads as: "The two species in this study exhibited different photosynthetic responses to seawater acidification, potentially due to their distinct inorganic carbon utilization strategies. The biochemical $CO_2$ concentrating mechanism (i.e. unicellular $C_4$ pathway) has been suggested to play a key role in the plasticity of inorganic carbon utilization in *T. weissflogii*

(Reinfelder et al., 2000). This mechanism is crucial for maintaining relatively high photosynthetic rates in responses to changes in pH. Additionally, differences in membrane permeability and fluidity may contribute to these species' varied responses. FAs are incorporated into phospholipids, which are major structural components of cell membranes, and FA composition can influence membrane characteristics. Theoretically, the cell membranes of *T. weissflogii* are less fluid and less permeable to $CO_2$ (Maulucci et al., 2016), due to higher proportions of SFA (Fig. 6). This helps cells cope with decreased intracellular pH and maintain homeostasis under seawater acidification conditions (Rossoll et al., 2012)." (Line 233-241)

How might the 10x higher rate of photosynthesis in T weissflogii influence your results?

Response: Are you suggesting that *T. weissflogii* cells have a photosynthetic rate that is 10 times higher than *S. costatum*? The rates shown in Figure 5 are normalized per cell. Two species have comparable chl a-normalized photosynthetic rates, as the chlorophyll a content of *T. weissflogii* is also 8-10 times the content of *S. costatum*. Thank you for pointing this out. We have discussed these interesting results in this section. (Line 242-248)

207: I think you should use "ambient and low pH levels" to be consistent with the rest of the paper. See my comment about stats in between panels in Figure 5 (above).

Response: We agree and use consistent terminology throughout the revised manuscript. For Figure 5, we integrate LC and HC panels of each species into a single panel and include the photosynthetic rate data of steady treatment from two measurements.

209: I like that you discuss the implication of your experiments, but this was too big of a jump. Please describe more about how your results suggest this overestimation problem.

Response: Thank you for your valuable suggestion. We revise these sentences to emphasize the differences between steady and fluctuating conditions and their influences on the predictions regarding the consequences of acidification as suggested. We also summarize effects of steady and fluctuating acidification on phytoplankton in Table 1. (Line 253-260)

212: What can your two test species tell you about variabilty across more species? What can you generalize about?

Response: Theoretically, coastal species should exhibit tolerance to fluctuating pH. However, it may not be rational to extrapolate the findings of the present study to other species. Your comments prompt us to summarize relevant published data to identify any potential general trends. We have collected studies on fluctuating pH and present main results in Table 1. Related discussions are also included. (Line 253-260)

223: Do these PUFA proportions match previous work?

Response: Yes, the proportions of EPA and DHA in two species fall within reported ranges for microalgae and diatoms. However, culturing conditions, especially stressful ones, can significantly alter fatty acid composition. PUFA proportion in a species can vary considerably under different conditions. We include additional information and relevant references to support this in the revised manuscript. (Line 270-273)

236: Why do two Thalassiosira have diverging PUFA responses to high pCO2??

Response: As mentioned above, culturing conditions play a crucial role in regulating FA compositions in microalgae. Stressful conditions are often used to induce lipid accumulation and PUFA synthesis. The pCO2 levels used in ocean acidification studies are typically not stressful for most species, which can resulting in various results for FA compositions and photosynthetic physiology. In addition, previous studies have reported species- or even strain-specific responses to CO2, likely related to differences in pH tolerance and efficiency of CO2 concentrating mechanisms. It has been proposed that pH might act as a regulation signal for the formation of cell membranes, which are mainly composed of fatty acids, by controlling the production of its synthesizing enzymes. In response to your and the other referee's suggestions, we include discussions on the possible underlying mechanisms that drive changes in FA compositions. (Line 283-288)

**Technical corrections:**

Line 10:   add "the", …."Research on the influences…"

Line 11: ecosystems plural

Line 17: lead to

Line 18: coastal

Line 28: the dominant

Line 28: where they contribute to a large

Line 34: given that the pH difference

Line 36: dwelling

Line 44: hinders

Line 45: the impacts

Line 47: enabled

Line 62: Add (see Fig. 1) at end of sentence.

Line 139: change "at each pH level" to "in   LC or HC conditions".

Line 171: remove "was"

Line 183: "which is larger than the average expected change of 0.3 units…

Line 185: "of large pH fluctuations" in coastal waters…

Line 186: There "are limited studies" investigating….

Line 187: , "and typically neutral or positive effects are observed"

197: which one?

198: delete "even" , replace place with "and", insert "even" after was.

199: regimes, plural

204: add figure reference (Fig 5e,f)

206: differences, plural

225: delete "Kinds"

Response: We appreciate your detailed corrections! We thoroughly check the manuscript to address these typos and vocabulary issues, and we have included supplementary information in the revised manuscript.